# An Application of Metabolic Syndrome Severity Scores in the Lifestyle Risk Assessment of Taiwanese Adults

**DOI:** 10.3390/ijerph17103348

**Published:** 2020-05-12

**Authors:** Chih-Ming Lin

**Affiliations:** Department of Healthcare Information and Management, Ming Chuan University, Taoyuan 333, Taiwan; cmlin@mail.mcu.edu.tw; Tel.: +886-3-350-7001 (ext. 3530)

**Keywords:** cardiovascular disease, confirmatory factor analysis, ethnicities, lifestyle, metabolic syndrome

## Abstract

A metabolic syndrome (MS) diagnosis was made when the criteria for three or more of five MS components were met. Due to some limitations in the traditional MS criteria, however, different health care societies have sought to develop applicable MS scoring systems instead. Continuous MS scores can be of meaningful value in the prevention, diagnosis, and treatment of MS at different life stages. Relatedly, this study used a database for 27,748 subjects aged 20 to 64 years who received health checks at a health screening institution in Taiwan from 2010 to 2015 to a similar end. Five components of MS (waist circumference, fasting plasma glucose, blood pressure, fasting triglycerides, and high-density lipoprotein) were used to formulate an MS severity score in different gender and age stratums, which was then used to evaluate the risks of various lifestyle habits. Those estimates were then compared with the results for traditional MS diagnosis. The MS severity scores for some behaviors relating to smoking, drinking, physical activity, and sweetened beverage consumption were found to have changed from 0.03 to 0.2; however, a logistic regression analysis with dichotomous diagnosis did not indicate significant links between these behaviors and MS. The models established by the MS severity scores can identify the risk factors for MS in a more sensitive manner than the traditional MS diagnosis can, especially with respect to specific lifestyle habits. MS severity score can serve as an indicator to explore the potential risk factors for subclinical conditions in the early stages of MS.

## 1. Introduction

Metabolic syndrome (MS) is a risk factor for type 2 diabetes and cardiovascular diseases (CVD), and it is also associated with various burdens placed on health care systems. As obesity rates rise and population aging continues, the prevalence of MS is also rising. A recent study showed that the prevalence rates of MS in China (21.3%) and South Korea (31.3%) have grown by about 30% over the past 10 years, and nearly doubled in Taiwan over the past 12 years, from 13.6% to 25.5% [1]. A China-based study reported that this trend may have been caused by elevated uric acid levels [2]. Nevertheless, MS can be prevented by assessing and addressing of risk factors, such as personal socio-economic conditions and lifestyle behaviors, which include smoking, alcohol consumption, dietary habits, physical activity [3,4]; the intake of antioxidants (e.g., vitamin C and vitamin E) [4,5,6,7]; personal income; occupation; and education [8,9,10].

There has been a question regarding the clinical significance of MS. More specifically, since the five MS components (i.e., waist circumference, fasting plasma glucose (FPG), blood pressure, fasting triglycerides (TG), and high-density lipoprotein (HDL)) are considered CVD risk factors, views still differ on whether another integrated criterion is actually helpful. Gale has argued that MS is a redundant diagnosis for people with diabetes, and recommends that the criteria for determining MS should exclude patients with diabetes and CVD [11]. Meanwhile, past studies have found that the continuous/numerical score is more sensitive and less likely to be erroneous when used in statistical analysis than the dichotomous classification [12,13]. Kahn et al. pointed out that, due to information gaps in the current definition of MS, clinicians should assess and treat all relevant risk factors for CVD, rather than only considering the diagnostic criteria for MS. They further suggested that a continuous MS measure could be more efficient for disease prevention [14]. Previous studies on the scoring of continuous MS measures have used several tools such as principle component analysis [15,16], Z scores [17,18], percentile rankings [19], and factor analysis [20]. Age can be a crucial factor for predicting the development of CVD and needs to be included into any valid scoring system. Relatedly, Eisenmann argued that it is necessary to use the standardized MS score to estimate MS risk due to diabetes, atherosclerosis, and CVD from childhood to adulthood [21]. Continuous MS scores can thus be of meaningful value in the prevention, diagnosis, and treatment of MS at different life stages.

Continuous MS scores have recently been developed for Asian adults. Jung et al. found a high correlation between continuous MS scores and both ischemic stroke and heart disease mortality [22]. Two recent studies developed MS severity scores for Asian populations (i.e., Koreans, Chinese, Malaysians, and Indians) to assess the risk of the development of diabetes and inflammation markers [23,24]. However, few studies have linked continuous MS scores to socioeconomic and lifestyle conditions, as well as to Taiwanese populations. The present study was thus aimed at developing gender- and age-specific MS severity scores for Taiwanese adults, and at assessing the associations between individual lifestyle habits and those severity scores in comparison with the traditional MS diagnosis.

## 2. Methods

### 2.1. Data Source

The study collected and analyzed data from the Major Health Screening Center in Taiwan. The center is a membership-oriented private institution with four clinics located around the country that provide periodic health examinations to the center’s members. Each member participated in a check-up program that offers a discounted examination fee for receiving the examination repeatedly over multiple years. The data collection and analysis of the resulting Major Longitudinal Health-Check-Up-Based Population Database (MJLPD) has been described in previous reports [25,26]. The MJLPD database is made accessible to academic researchers upon request. As various ethical issues could arise from data usage, the protocol of this study was evaluated and agreed to by the Research Ethics Committee of National Taiwan University (NTU-REC 201911ES012) and the Major Health Screening Center.

### 2.2. Study Sample

In the study, we collected data from 71,108 participants aged 20 to 64 years who underwent their first standard health screening at the center from 2010–2015. 11,093 participants with CVD including heart diseases, stroke, and diabetes or those receiving related treatments were excluded from the study, since these conditions could produce perturbations in MS-related measures that might mask potentially important relationships. Furthermore, to minimize selection bias, study subjects were calibrated and selected randomly from the remaining participants to reflect the sex and age composition of Taiwan’s population. A final total of 27,748 subjects (13,823 males and 13,925 females) met the inclusion criteria for analysis.

### 2.3. Response Variables

In our study, MS was regarded as a dichotomous variable and defined according to the modified Adult Treatment Panel III (ATP III) criteria and the official criteria announced by Taiwan’s National Health Promotion Administration [27]. A MS diagnosis was made when 3 or more of the following conditions were present: Waist circumference ≥ 90 cm in men and ≥ 80 cm in women; fasting plasma glucose (FPG) ≥ 100 mg/dL (5.55 mmol/L) or use of antidiabetic medication; systolic blood pressure (SBP) ≥ 130 mmHg, diastolic blood pressure ≥ 85 mmHg, or use of antihypertensive medication; fasting triglycerides (TG) ≥ 150 mg/dL; and high-density lipoprotein (HDL) cholesterol < 40 mg/dL in men and < 50 mg/dL in women.

The general methods for formulating MS severity scores have been reported previously [20,28]. Confirmatory factor analysis was performed for the data of adults aged 20–64 years who were categorized into six subgroups based on gender and the following age groups: 20–34, 35–49, and 50–64 years. Those MS severity scores were then presented as scores (i.e., scores that range from theoretical negative to positive infinity and have a mean of 0 and standard deviation (SD) of 1, as well as a normal distribution) of relative MS severity for a given gender and age range.

### 2.4. Explanatory Variables

The study subjects had each completed a self-administered questionnaire during screening that provided information on their sociodemographic characteristics and lifestyle habits. In addition to sex and age, we collected data regarding four aspects of socioeconomic status (i.e., marital status, education, income, and occupation) and twelve lifestyle habits including smoking, alcohol consumption, betel nut chewing, physical activity (i.e., duration, intensity, and frequency), sleep habits, vegetarian diet, drinking sweetened beverages, and taking nutritional supplements (i.e., vitamin C/E and fish oil), which are all well-documented as constituting related risk factors. Sex, age, and socioeconomic status played the role of confounders in the multi-variable analysis.

### 2.5. Statistical Analysis

Descriptive statistics regarding the above characteristics were calculated for all participants. The prevalence of MS and average MS severity scores were calculated across the gender and age groups. A confirmatory factor analysis approach was used to derive the MS severity scores based on the five MS components, with a weighted contribution for each of the components to a latent MS factor being determined on the basis of both specific age ranges and genders [29]. SBP, rather than diastolic blood pressure, was chosen for this factor analysis because SBP has a stronger association with insulin resistance [30,31]. Furthermore, log-transformed values of TG levels were used because the TG levels of the collected data exhibited a skewed distribution. Meanwhile, in order to be interpretable in a similar manner as the other measures included in the model, inverse values of the HDL cholesterol were utilized. The individual values of five components were then standardized and converted to Z scores before the factor analysis was performed.

We wanted to determine the factor loadings because each one indicated the magnitude of the corresponding association between an associated component and the underlying MS factor. Several fit indices such as the Chi square, Akaike information criterion, root mean square error of approximation, standardized root mean square residual, goodness of fit index, and Bentler-Bonett normed fit index were used to estimate the parameters of the confirmatory factor analyses in accordance with the related criteria [28]. For each of the six subgroups, factor loadings for the five MS components were determined based on one factor (i.e., MS latent factor). Models were constructed, and then selected according to the fit measure indices. Our results showed the factor loadings estimated by the optimal model. The process of factor analysis has been described with formulas in a study conducted by Low et al. [24]. We also calculated MS severity scores based on unstandardized values for the traditional five components in order to provide scores that could be used relatively easily in clinical contexts. These scores were derived by back-transforming the standardized coefficients and a covariance matrix obtained from the factor analysis. The relationships between the dichotomous MS results (that is, whether an individual has or does not have MS), the quartile-ordinal outcomes of the MS severity scores, and the MS severity scores and risk factors were likewise examined by performing a binary logistic regression, ordinal regression (i.e., cumulative logistic regression), and linear regression model, respectively. The confirmatory factor analysis and regression analyses were performed using R, AMOS, and SPSS (version 22.0; IBM Institute Inc., New York, NY, USA).

## 3. Results

The overall prevalence rates of MS in this sample were about 16.7% in men and 7.0% in women according to the ATP III criteria (Table 1). Women had a lower prevalence of MS, and the prevalence of MS was clearly increased in those more than 50 years old. Upon performing the confirmatory factor analysis of the MS components, we generated models that were well fit to the data (Table 2), with the overall models exhibiting acceptable model fit. Regarding factor loadings, there were notable age- and gender-related differences in the magnitudes of the factor loadings. Among the MS components, waist circumference had the highest factor loadings, indicating that it held the strongest correlation with MS among the investigated sample. SBP and FPG had the lowest factor loadings in men aged 35 to 64 years and women aged 50 to 64 years. Equations based on the factor coefficients from the confirmatory factor analysis results are presented in Table 3. Individual MS severity score can be calculated with the established equation. The mean of the MS severity scores of those with MS and those without MS was 1.259 and –0.167, respectively. Observing the distributions of the MS severity scores shown in Figure 1, majority of the participants with MS had a severity score of more than zero, and a medium value of 1 and 1.5 for men and women. However, the distributions of the scores for those with MS had major overlaps with the scores for those without MS. The classification cannot be observed clearly until the severity scores are greater than two.

Table 4 shows the prevalence and severity scores of MS for different lifestyle habits. Unhealthy habits such as smoking, drinking alcohol, chewing betel nut, and drinking sweetened beverages represent higher MS risks, while getting sufficient sleep, engaging in physical activity, and taking vitamin C supplements have positive effects on health. The MS severity scores indicated a more obvious link to lifestyle habits, especially for physical activity and drinking sweetened beverages, and this link was not observed in the dichotomous diagnosis.

Using three regression models, the risk factors for MS or its severity were estimated (Table 5). With the MS severity score as the outcome variable, it was observed that persons who smoked had a 11–13% increased risk of MS, or a severity score of 0.07, compared to non-smokers. Drinking alcohol more than 3 times a week caused a 17–25% decrease in MS risk, or a severity score of 0.12–0.19. Drinking sweetened beverages week resulted in more than 0.03 severity score. Unlike the findings with the MS severity score, the three lifestyle habits were found to have no relation to MS in a logistic regression analysis with the dichotomous diagnosis. Similar observations were made for physical activity frequency; that is, regular exercise was found to protect against MS when using the model with the severity score, but the benefit could not be observed in persons who exercised when using the model with the dichotomous diagnosis. Betel nut intake was also found to have an adverse effect on MS when a logistic model was used, but the effect could not be found when using a linear model. Irrespective of the model used, meanwhile, our multi-variable analysis showed that sleep, duration of physical activity, vegetarian diet, and taking vitamin E supplements were not related to the occurrence of MS or its severity. Furthermore, vitamin C and fish oil intake were found to have opposite effects on MS severity when using a linear model.

## 4. Discussions

The present study calculated MS severity scores for Taiwanese adults (other than elderly adults) that take variations in how MS is manifested in different gender and age groups into account by placing differential weights on individual MS components. Waist circumference and TG levels were found to dominate the contributions of the factor loadings for MS. HDL cholesterol levels had particularly high factor loadings among older women. SBP exhibited lower factor loadings than the other components, which corresponds to the findings of previous studies [23,28]. The majority of the participants with MS had a positive score, which implies that zero may be a critical cut-off point for evaluating MS severity. Moreover, the large overlap in the score distribution of those diagnosed with MS and those not diagnosed with MS might mean that there are lots of individuals who do not suffer from MS but who have high severity scores due to having been neglected by clinics for potential risk of CVD. We also revealed that the MS severity scores in our study were highly correlated with lifestyle habits—including smoking, drinking alcohol, consuming sweetened beverages, and physical activity—which was not the case, however, for traditional MS diagnosis. To our knowledge, this is the first study to investigate an age- and gender-specific MS scoring system for Taiwanese adults based on the contribution of individual MS components, and to apply the system to the underlying risk factors.

Some confusion may arise from the traditional ATP III criteria, such as whether individuals with two high level MS components have lower CVD risk than those whose levels are slightly above the criteria in three or more components. By focusing on the purpose of health education, physicians can provide advice on disease prevention based on the risk severity levels for different groups and, consequently, promote public health [29,30]. The World Health Organization has stated that MS may not be suitable for clinical diagnosis although new community-based prevention strategies should nonetheless be developed and evaluated [31]. In addition, as age plays an important effector role in CVD, the traditional MS classification criteria may be questionable given their failure to consider age [30,32]. Eisenmann has argued, meanwhile, that a continuous MS score that is estimated from a specific population cannot be generalized. In other words, the MS score formulas used for different ethnic, sex, and age subgroups should be different [21]. Thus, estimations of MS severity or CVD risk should be established and calibrated based on specific populations. The present study provided a solution that addresses the question regarding the clinical significance of MS, while also evaluating the risk factors of MS. In the future, relationship verification and personal calculator development could be carried out used the established sex-age equation, so as to apply the MS severity scores of individuals in a clinical context and assess temporal changes in cardiometabolic risk.

Previous studies that estimated the Z scores of MS components for men and women showed that such a method can be helpful in making clinical assessments of MS correlations between parents and children, and could even be used to develop applications that can used on a calculator [33,34]. Guseman et al. noted that most studies have used cross-sectional designs and neglected to consider age; however, there is insufficient evidence to prove whether or not high risk scores in adolescence are related to CVD risk in adulthood, so it is necessary to develop indicators which could be applied to longitudinal estimation during different life stages [33]. Using the MS risk scoring system developed by Sullivan et al. and adjusting it with age factors in a Chinese population, Kang et al. created a recipient’s operational characteristic curve estimation of CVD risk, and found that continuous MS scores are more accurate in predicting CVD risk. They also found that the continuous MS score is not linearly related to CVD risk, and suggested that future scores should be developed for different racial, sex, and age subgroups [35]. Our findings revealed that the use of ordinal or linear regression analysis combined with sex- and age-specific MS severity scores is more sensitive in terms of identifying the related risk factors, a revelation which provides some empirical references for future research.

Another well-known continuous indicator for predicting CVD other than MS scores is the Framingham risk score (FRS), which considers age and can predict CVD risk for the following 10 years. A person is considered to have relatively low risk if the FRS is less than 10%. Nevertheless, due to lack of estimation of obesity and blood glucose levels, some studies have suggested that the FRS has limitations in the development of disease prediction and prevention strategies for different age groups [36,37]. A study investigating an Asian-Indian population noted that using the FRS to estimate CVD risk for Asian populations could result in underestimations, which might be present mainly in children and adolescents and MS patients [38]. Using the same data source as the present study, Liao and Lin recently determined FRS values for patients diagnosed with MS but without diabetes or other CVD, and found that these relatively healthy MS cases had fairly low FRS results (with an average of only 1.4%) [25]. Thus, a relatively low FRS could be easily ignored by physicians or patients and thus be ineffective for the purpose of early disease prevention.

The present study has a credible study design. In addition to exploring potential risk factors by analyzing a large sample database, this study also established a continuous MS indicator stratified by gender and age for a specific Asian population. Nevertheless, there are also limitations that need to be considered. As this is a cross-sectional study, its analyses cannot be used to establish causal relationships. Additionally, some independent variables were missing, which may diminish the validity of the study slightly. Moreover, its participants were healthier than the general population because those who had CVD or took related medicines and the elderly were excluded. As such, the risk estimations cannot be suitable for adults other than those who met the selection criteria.

## 5. Conclusions

Not intending to replace the function of traditional MS diagnosis, this study adds evidence for wide application of MS severity scores among an Asian population. Due to being more sensitive than traditional MS diagnosis when used to predict MS risks associated with various lifestyle habits, the use of MS severity scores can be promoted among individuals or health institutions for disease prevention. Health caregivers can utilize MS severity scores to propose or assess lifestyle-related prevention strategies for those at risk of underlying disease by monitoring the fluctuations in their MS severity scores over time.

## Figures and Tables

**Figure 1 ijerph-17-03348-f001:**
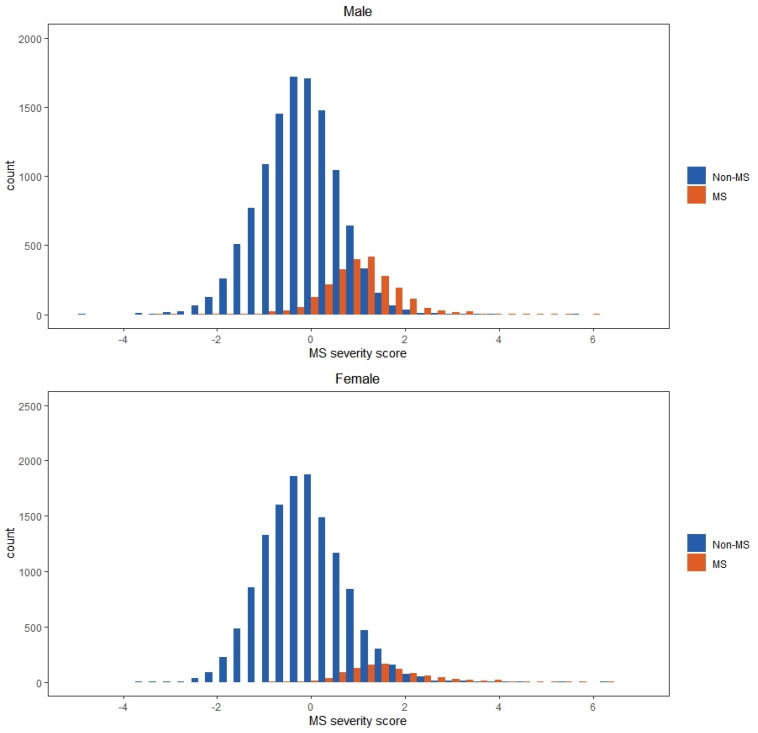
Distributions of MS severity scores in men and women.

**Table 1 ijerph-17-03348-t001:** Metabolic syndrome and its components for sex- and age-specific groups

Groups	*T*	MS (%)	WC		FPG		TG		HDL-C		SBP	
			Mean	SD	Mean	SD	Mean	SD	Mean	SD	Mean	SD
Men	13823	16.7	82.54	8.39	101.35	14.09	124.46	76.2	53.29	12.05	119.91	13.79
20–34 YRS	4816	11.3	81.37	9.17	97.8	10.29	109.26	69.93	53.99	12.02	119.39	12.52
35–49 YRS	4803	18.7	83.1	8.02	101.51	13.61	136.19	79.78	51.87	11.33	119.15	13.34
50–64 YRS	4204	20.6	83.23	7.7	105.23	17.05	128.46	76.03	54.11	12.72	121.36	15.46
Women	13925	7	70.85	7.39	96.14	12.32	87.04	52.89	66.42	15.21	109.49	15.07
20–34 YRS	4583	2.1	68.18	7.14	92.42	8.28	71.57	37.28	67.42	15.09	104.59	11.53
35–49 YRS	5073	5.8	70.59	6.69	95.53	9.76	85.19	53.63	65.41	14.78	107.91	13.62
50–64 YRS	4269	14.2	74.02	7.22	100.84	16.46	105.85	60.02	66.55	15.76	116.63	17.26

Abbreviations: YRS = years, MS = metabolic syndrome; SD = standard deviation; WC = waist circumference (in centimeters), FPG = fasting plasma glucose (in mg/dL), TG = triglycerides (in mg/dL), HDL-C = high-density lipoprotein cholesterol (in mg/dL); SBP = systolic blood pressure (in mmHg).

**Table 2 ijerph-17-03348-t002:** Model fit indices and factor loadings used in the confirmatory factor analysis of sex- and age-specific groups

Indices/Loadings	Men			Women		
	20–34 YRS	35–49 YRS	50–64 YRS	20–34 YRS	35–49 YRS	50–64 YRS
Indices						
Chi-square	85.353	96.288	88.174	23.355	65.790	31.043
AIC	107.353	118.288	110.174	45.355	87.790	53.043
RMSEA	0.065	0.069	0.071	0.032	0.055	0.040
SRMR	0.027	0.029	0.033	0.015	0.023	0.016
GF1	0.993	0.992	0.992	0.998	0.995	0.997
NFI	0.947	0.962	0.958	0.988	0.978	0.988
Factor loading						
WC	0.82	0.76	0.81	0.74	0.73	0.70
FPG	0.35	0.28	0.31	0.39	0.46	0.38
Ln-TG	0.52	0.49	0.42	0.44	0.50	0.50
HDL-C	0.40	0.36	0.38	0.35	0.38	0.41
SBP	0.41	0.35	0.24	0.38	0.37	0.36

Abbreviations: YSR = years, AIC = Akaike information criterion, RMSEA = root mean square error of approximation, SRMR = standardized root mean square residual, GFI = goodness of fit index, NFI = Bentler-Bonett normed fit index. Note: GFI, NFI < 0.90 or RMSEA, SRMR > 0.08 indicates a poor fit. Factor loadings > 0.3 were considered clinically meaningful.

**Table 3 ijerph-17-03348-t003:** Equations used to calculate sex- and age-specific metabolic syndrome severity scores

Sex-age Groups	Equation
Men	
20–34 YRS	−10.6959 + 0.0844 × WC + 0.0119 × FPG + 0.3680 × Ln-TG − 0.0082 × HDL-C + 0.0121 × SBP
35–49 YRS	−12.1104 + 0.0960 × WC + 0.0094 × FPG + 0.4556 × Ln-TG − 0.0097 × HDL-C + 0.0127 × SBP
50–64 YRS	−11.3783 + 0.1089 × WC + 0.0073 × FPG + 0.2835 × Ln-TG − 0.0086 × HDL-C + 0.0055 × SBP
Women	
20–34 YRS	−12.6514 + 0.1032 × WC + 0.0253 × FPG + 0.5074 × Ln-TG − 0.0100 × HDL-C + 0.0175 × SBP
35–49 YRS	−12.3220 + 0.0972 × WC + 0.0246 × FPG + 0.5251 × Ln-TG − 0.0089 × HDL-C + 0.0131 × SBP
50–64 YRS	−11.1397 + 0.0902 × WC + 0.0127 × FPG + 0.5491 × Ln-TG − 0.0093 × HDL-C + 0.0112 × SBP

MS severity scores are generated by inserting an individual’s clinically measured values for WC (in centimeters), FPG (in mg/dL), Ln-TG (in mg/dL), HDL-C (in mg/dL), and SBP (in mmHg); YRS = years.

**Table 4 ijerph-17-03348-t004:** MS prevalence and severity scores categorized according to lifestyle habits

Lifestyle Habit	MS		MS Severity Score		Total
	*n*	%	Mean (SD)	Median	
Smoking					
None	1939	9.9%	−0.024 (0.974)	−0.079	19,590
Second-hand smoke	118	11.0%	0.058 (1.126)	−0.044	1069
Quit	281	17.0%	0.038 (0.970)	−0.011	1652
Casual intake	135	13.8%	0.017 (1.003)	−0.032	978
Daily intake	641	18.3%	0.080 (1.015)	0.032	3495
Missing data	168	17.4%	0.108 (1.301)	0.038	964
Drinking					
None	2297	10.7%	−0.004 (0.987)	−0.066	21,384
Quit	67	13.3%	0.027 (0.973)	0.008	502
1–2 times/wk	400	15.3%	−0.001 (0.982)	−0.039	2608
3–4 times/wk	150	17.5%	−0.057 (1.002)	−0.068	858
>4 times/wk	58	19.1%	−0.026 (0.957)	−0.054	304
Missing data	310	14.8%	0.082 (1.154)	0.025	2092
Chewing betel nut					
None	2728	10.9%	−0.015 (0.981)	−0.067	25,105
Quit	24	40.7%	0.166 (1.043)	0.105	840
1–2 times/wk	70	27.9%	0.352 (1.199)	0.341	251
3–4 times/wk	194	23.1%	0.567 (1.045)	0.612	59
>4 times/wk	24	25.8%	0.189 (0.978)	0.026	93
Missing data	242	17.3%	0.096(1.215)	0.003	1400
Sleeping (hrs/day)					
<4	45	13.8%	0.082 (1.052)	−0.034	327
4.0–5.9	719	12.1%	0.029 (1.010)	−0.034	5959
6.0–6.9	1592	12.0%	−0.012 (0.974)	−0.059	13,267
7.0–7.9	681	10.7%	−0.016 (0.985)	−0.063	6349
≥8	104	10.1%	−0.029 (1.004)	−0.113	1031
Missing data	141	17.3%	0.165 (1.353)	0.078	815
Physical activity (level)					
None	2311	11.9%	0.030 (0.998)	−0.024	19,344
Light	588	11.1%	−0.069 (0.937)	−0.120	5313
Moderate	120	8.4%	−0.230 (0.897)	−0.259	1426
Heavy	42	11.6%	0.039 (1.066)	−0.031	363
Missing data	221	17.0%	0.109 (1.277)	0.006	1302
Physical activity (times/wk)					
None	881	12.0%	−0.011 (1.038)	−0.080	7340
1	497	11.9%	−0.059 (0.950)	−0.086	4189
2–3	644	10.7%	−0.057 (0.960)	−0.104	6003
7	812	11.2%	0.009 (1.016)	−0.054	7265
>7	129	12.9%	0.049 (0.978)	−0.011	997
Missing data	319	16.3%	0.117 (1.187)	0.027	1954
Physical activity (hrs/day)					
<0.5	948	11.4%	0.027 (0.993)	−0.031	8305
0.5–1	1162	11.4%	−0.006 (0.991)	−0.052	10,198
1–2	576	10.9%	−0.055 (0.964)	−0.118	5267
>2	223	13.3%	−0.021 (0.959)	−0.053	1679
Missing data	373	16.2%	0.093 (1.156)	−0.009	2299
Vegetarian diet					
No	3145	11.7%	−0.002 (0.989)	−0.057	26,804
Yes	98	12.8%	0.043 (1.085)	−0.026	766
Missing data	39	21.9%	0.389 (1.794)	0.348	178
Drinking sweetened beverages (cups/wk)					
None	1236	12.2%	−0.023 (0.986)	−0.066	10,167
1–3	364	10.5%	0.017 (0.991)	−0.043	3464
4–6	403	10.8%	0.020 (0.942)	−0.031	3727
7	999	11.7%	−0.022 (0.969)	−0.070	8508
>7	172	13.7%	0.044 (1.150)	−0.069	1260
Missing data	108	17.4%	0.142 (1.418)	0.007	622
Taking vitamin C supplements					
No	2927	12.2%	0.013 (1.004)	−0.042	23,983
Yes	355	9.4%	−0.073 (0.967)	−0.116	3765
Taking vitamin E supplements					
No	3096	11.9%	0.006 (1.003)	−0.049	26,121
Yes	186	11.4%	−0.058 (0.951)	−0.105	1627
Taking fish oil supplements					
No	3063	11.7%	0.001 (0.999)	−0.055	26,172
Yes	219	13.9%	0.037 (1.010)	−0.019	1576

Abbreviations: MS = metabolic syndrome; SD = standard deviation.

**Table 5 ijerph-17-03348-t005:** Risk factors of lifestyle habits for metabolic syndrome in different regression models

Lifestyle Habit	Logistic Regression		Ordinal Regression		Linear Regression	
	AOR	*p*-Value	AOR	*p*-Value	β	*p*-Value
Smoking (vs. None)						
Second-hand smoke	1.100	0.452	1.002	0.782	0.060	0.113
Quit	1.104	0.684	1.132	0.027	0.071	0.022
Intake casually	1.077	0.731	1.110	0.129	0.065	0.092
Intake everyday	1.136	0.505	1.112	0.019	0.068	0.007
Drink (vs. None)						
Quit	0.819	0.228	0.832	0.062	−0.122	0.027
1–2 times/wk	0.951	0.502	0.914	0.045	−0.048	0.053
3–4 times/wk	0.956	0.757	0.830	0.013	−0.134	0.001
>4 times/wk	0.854	0.414	0.752	0.023	−0.191	0.008
Chewing betel nut (vs. None)						
Quit	1.272	0.033	1.106	0.213	−0.163	0.250
1–2 times/wk	1.859	0.001	1.561	0.002	0.083	0.061
3–4 times/wk	2.973	0.004	3.216	0.001	0.122	0.437
>4 times/wk	1.060	0.869	1.375	0.208	0.323	0.165
Sleep (vs. ≥8hrs/day)						
<4hrs/day	1.034	0.888	1.141	0.348	0.025	0.761
4.0–5.9 hrs/day	0.959	0.759	1.108	0.153	0.027	0.692
6.0–6.9 hrs/day	0.982	0.891	1.107	0.325	0.038	0.114
7.0–7.9 hrs/day	0.896	0.416	1.105	0.491	−0.014	0.672
Physical activity level (vs. None)						
Light	0.792	<0.001	0.837	<0.001	−0.088	<0.001
Moderate	0.572	<0.001	0.653	<0.001	−0.236	<0.001
Heavy	0.669	0.251	0.742	0.066	−0.221	0.016
Physical activity frequency (vs. None)						
1 time/wk	0.793	0.075	0.935	0.366	−0.003	0.873
2–3 times/wk	0.853	0.210	0.876	0.003	−0.038	0.091
7 times/wk	0.899	0.395	0.940	0.125	−0.085	0.001
>7 times/wk	0.977	0.853	0.988	0.744	−0.055	0.178
Physical activity duration (vs. <0.5 hrs/day)						
0.5–1 hrs/day	1.044	0.491	1.027	0.439	0.024	0.453
1–2 hrs/day	0.966	0.649	0.914	0.355	0.038	0.347
>2 hrs/day	0.935	0.511	0.943	0.317	−0.014	0.490
Vegetarian diet (vs. No)						
Yes	1.003	0.983	0.989	0.889	−0.007	0.800
Drinking sweetened beverages (vs. None)						
1–3 cups/wk	1.115	0.057	1.024	0.450	0.038	0.032
4–6 cups/wk	1.004	0.961	1.060	0.150	0.032	0.146
7 cups/wk	0.983	0.822	0.984	0.704	−0.006	0.780
>7 cups/wk	1.080	0.479	0.985	0.813	0.052	0.037
Taking vitamin C supplements (vs. No)						
Yes	0.849	0.036	0.883	0.001	−0.067	0.002
Taking vitamin E supplements (vs. No)						
Yes	1.046	0.671	0.945	0.332	−0.039	0.225
Taking fish oil supplements (vs. No)						
Yes	1.204	0.048	1.095	0.110	0.071	0.022

Adjusted odds ratios (AOR) and β are calculated by three regression models adjusted with all covariates, including demographic variables and socioeconomic status. Logistic regressions are calculated with ATP III criteria for MS. Linear regression and ordinal logistic regressions are calculated with MS severity score and its quartile outcome.

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
