# Peer review of "An Application of Metabolic Syndrome Severity Scores in the Lifestyle Risk Assessment of Taiwanese Adults"

_ijerph, 2020, doi:10.3390/ijerph17103348_

Round 1

Reviewer 1 Report

The study aimed to develop gender- and age-specific equations to determine metabolic syndrome (MS) severity score for Taiwanese adults. The study also aimed to assess associations between lifestyle and severity scores compared with the traditional MS diagnosis.

The manuscript is generally well written and clear. Also, considering its large sample size, the findings from the present study may be valuable for Taiwanese adults. However, there are few points that need clarification.

- Figure 1. The results showed large overlaps in the MS severity scores between participants with or without MS regardless of genders. Is it possible to provide a mean scores for the MS severity scores observed from those with MS and without MS and also to show specificity and sensitivity?

- There are several typo errors such as errors in font type and a space between a reference and the word in front of the reference. Please check.

Author Response

1.Figure 1. The results showed large overlaps in the MS severity scores between participants with or without MS regardless of genders. Is it possible to provide a mean scores for the MS severity scores observed from those with MS and without MS and also to show specificity and sensitivity?

Answer:

The mean scores for the MS severity scores observed from those with MS and without MS are 2.2257 and -0.2968, respectively. (Lines 148-150) The purpose of the study is not to predict MS with our methods. Otherwise, we noticed the question regarding the clinical significance of MS and tried to provide a solution to deal with the problem while evaluating the risk factors of MS. We are not able to evaluate accuracy for each person due to his unknown value of MS score. Instead, we evaluated overall performance of our model across all samples by goodness-of-fit indices commonly used in the field of SEM. We add the statements in Discussion. (Lines 231-233)

2.There are several typo errors such as errors in font type and a space between a reference and the word in front of the reference. Please check.

Answer:

Thank you for your indication, the errors have be processed.

Reviewer 2 Report

Please see my comments below:

1) Please spell out MS in the abstract especially when it is used for the first time. This will be helpful for the lay audience. Also, please provide one sentence on the background of the study.

2) Lines 11: what do you meant by to a similar end?

3) abstract: please be specific – is this MS severity score developed for which age group etc?

4) English needs to be improved entirely throughout the manuscript because I can spot some grammatical error when even reading the abstract, eg. Line 15.

5) abstract: what are the criteria or score for traditional MS diagnosis? Please mention this in your methods in the abstract too. You need to let the readers to know how you came to such methods or results.

6) lines 15-16: your abstract didn't include any values or statistical analyses to indicate tht this is a more sensitive manner.

7) Abstract: suggest to revise the abstract, include some important results to support your conclusions.  

8) Introduction: One feature that is common to participants with metabolic syndrome is an elevated uric acid. Elevated uric acid is now also commonly seen in adolescents. Please include this reference as well: Zhou, H., Ma, Z. F., Lu, Y., Du, Y., Shao, J., Wang, L., Wu Q, Pan, B, Zhu, W & Wei, H. (2020). Elevated serum uric acid, hyperuricaemia and dietary patterns among adolescents in mainland China. Journal of Pediatric Endocrinology & Metabolism: JPEM. DOI: https://doi.org/10.1515/jpem-2019-0265

9) Line 32: what are the debates in this area? Please include some details. No citations are even provided.

10) Line 65: were there any links for this database?

11) Line 74: How was the sampling conducted? It is unclear here.

12) please include ethical approval no.

13) please include inclusion and exclusion criteria clearly. The points are scattered all over the paragraphs in method section and the readers will have difficulty to follow your points.

14) how was the randomisation performed? By who?

15) line 87: what are the 6 subsgroups?

16) line 110: why one-factor model is used? MS is caused by more than one factors.

17) lines 114-115: please provide the calculation of the MS score using the standardised factor coefficients from the model. Can provide the example of how to calculate the MS score of one participant.

18) lines 117-118: what tests are used for examining the linear relationships?
19) table 1: please include some p-values in table 1, using regression.

20) table 2: please include some discussion on the results to guide the readers to understand this table 2 better.

21) figure 1 can be removed. I don't see any meaningful interpretation derived from figure 1.

22) how was the missing data handled? Did the author perform a sensitivity analysis?

23) table 5: why there are 3 different logistic regression performed? Logistic, ordinal and linear regression? This is confusing.

24) Discussion: I see little value in this study. I think the study results needs to be re-interpreted and foucs on the major findings. For example, the equations derived-  were they tested for their validity and usefulness?

25) References: please revise and format them properly.

Author Response

 1)Please spell out MS in the abstract especially when it is used for the first time. This will be helpful for the lay audience. Also, please provide one sentence on the background of the study.

Answer:

Thank you for your indication, metabolic syndrome has been added according to your suggestion. (Line 8)

2)Lines 11: what do you meant by to a similar end?

Answer:

Thank you for your indication, the misinterpreted statement has been removed.

3)abstract: please be specific – is this MS severity score developed for which age group etc?

Answer:

The age group has been added. (Line 12)

4)English needs to be improved entirely throughout the manuscript because I can spot some grammatical error when even reading the abstract, eg. Line 15.

Answer:

English has be reedited entirely throughout the manuscript

5) abstract: what are the criteria or score for traditional MS diagnosis? Please mention this in your methods in the abstract too. You need to let the readers to know how you came to such methods or results.

Answer:

The statement has been added. (Line 8 )

6) lines 15-16: your abstract didn't include any values or statistical analyses to indicate this is a more sensitive manner.

Answer:

The statement has been added. (Lines 17-20)

7) Abstract: suggest to revise the abstract, include some important results to support your conclusions.  

Answer:

The abstract has been revised according to your suggestions.

8) Introduction: One feature that is common to participants with metabolic syndrome is an elevated uric acid. Elevated uric acid is now also commonly seen in adolescents. Please include this reference as well: Zhou, H., Ma, Z. F., Lu, Y., Du, Y., Shao, J., Wang, L., Wu Q, Pan, B, Zhu, W & Wei, H. (2020). Elevated serum uric acid, hyperuricaemia and dietary patterns among adolescents in mainland China. Journal of Pediatric Endocrinology & Metabolism: JPEM. DOI: https://doi.org/10.1515/jpem-2019-0265

Answer:

The reference has been included. (Line 34)

9) Line 32: what are the debates in this area? Please include some details. No citations are even provided.

Answer:

In addition to Ref 11in the Introduction, the debates are described mainly in Discussion section with Ref 11, 21, 32, 33, 34. Thus, we instead the word “debates” with “question”. (Line 39)

10) Line 65: were there any links for this database?

Answer:

The database can be released by academic application. Please access the link: http://www.mjhrf.org/main/page/release2/en/#release05.

11) Line 74: How was the sampling conducted? It is unclear here.

Answer:

We collected all the participants aged 20 to 64 years who underwent their first standard health screening at the Center from 2010-2015. 11,093 participants with CVD were excluded from the 71,108 participants. Using the sampling option in SPSS package, study subjects were calibrated and selected from the remaining 60,015 participants to reflect the sex and age composition of Taiwan’s population. We describe the method more clearly. (Lines 78-84)

12) please include ethical approval no.

Answer:

The ethical approval no is NTU-REC 201911ES012 and has been included. (Lines 74-75)

13) please include inclusion and exclusion criteria clearly. The points are scattered all over the paragraphs in method section and the readers will have difficulty to follow your points. how was the randomisation performed? By who?

Answer:

We collected all the participants aged 20 to 64 years who underwent their first standard health screening at the Center from 2010-2015. 11,093 participants with CVD were excluded from the 71,108 participants. Using the sampling option in SPSS package, study subjects were calibrated and selected from the remaining participants according to the sex and age proportions of Taiwan’s inhabitants. We describe the method more clearly. (Lines 78-84)

14) line 87: what are the 6 subsgroups?

Answer:

Analysis was performed for adults who were categorized into six subgroups based on gender and the three age groups: 20–34, 35-54, and 55–64 years. We describe the method more clearly. (Lines 96-98)

15) line 110: why one-factor model is used? MS is caused by more than one factors.

Answer:

Thank you for your indication. For each of these six subgroups, factor loadings for the five MS components were determined based on a single MS factor. We describe the method more clearly. The misinterpreted statement has been removed.

16) lines 114-115: please provide the calculation of the MS score using the standardised factor coefficients from the model. Can provide the example of how to calculate the MS score of one participant.

Answer:

The standardized factor coefficient had been shown as a factor loading in Table 2. The calculation of participant’s MS score can be found in Table 3. These MS scores were derived by back-transforming the coefficients obtained from the standardized factor analysis. (Lines 119-132) Then, for example, the score of men aged 20-34 can be calculated in equation “-17.3032+0.0894WC+0.0340FPG+1.0087Ln-TG -0.0333HDL-C +0.0327SBP” with their five original value of MS components.

17) lines 117-118: what tests are used for examining the linear relationships?

Answer:

The relationships between the MS severity scores and risk factors (dummy variables) were examined with a multiple linear regression model. (Lines 132-136)

19) table 1: please include some p-values in table 1, using regression.

Answer:

Table 1 features mainly the distribution of MS and its components for the study subjects. As the confounders, the demographic variables have been included in the multiple regression shown as a note in Table 5. To test the differences of MS or its components among the sex-ages subgroups is not the purpose for this study. Please agree with our original presentation.

20) table 2: please include some discussion on the results to guide the readers to understand this table 2 better.

Answer:

The fit indices include Chi square, Akaike information criterion, root mean square error of approximation, standardized root mean square residual, goodness of fit index, Bentler-Bonett normed fit index. To guide the readers to understand this Table 2 easily, the criteria of the model fit indices have been noted in Table 2 and the main text (Lines 123-127).

21) figure 1 can be removed. I don't see any meaningful interpretation derived from figure 1.

Answer:

Figure 1 presents a question and a possible solution regarding the clinical significance of MS as our statements in Introduction and Discussion, such as whether individuals with two high level MS components have lower CVD risk than those whose levels are slightly above the criteria in three or more components. So please agree us to remain the Figure.

22) how was the missing data handled? Did the author perform a sensitivity analysis?

Answer:

Part of independent variables exist missing. In the SPSS package, the missing value was excluded before modeling. We did not perform a sensitivity analysis. Thank you for your comments. We add the limitation in Discussion. (Lines 267-268)

23) table 5: why there are 3 different logistic regression performed? Logistic, ordinal and linear regression? This is confusing.

Answer:

To compare the differences of analyses between the MS estimated methods the study tended to apply, we performed the three regression models to deal with the outcome variables as a dichotomous, ordinal and continuous variable, respectively. The ordinal regression model is as known as cumulative logistic regression model. We added the term in the statement to make it clearer. (Lines 132-136)

24) Discussion: I see little value in this study. I think the study results needs to be re-interpreted and foucs on the major findings. For example, the equations derived-  were they tested for their validity and usefulness?

Answer:

We are not able to evaluate accuracy for each person due to his unknown value of MS score. Instead, we evaluated overall performance of our model across all samples by goodness-of-fit indices commonly used in the field of SEM. The present study provided a solution to deal with the question regarding the clinical significance of MS while evaluating the risk factors of MS. Due to being more sensitive than traditional MS diagnosis when used to predict MS risks associated with various lifestyle habits, the use of MS severity scores can be promoted among individuals or health institutions for disease prevention. Health caregivers can utilize MS severity scores to propose or assess lifestyle-related prevention strategies for those at risk of underlying disease by monitoring the fluctuations in their MS severity scores over time. In the future, relationship verification and personal calculator development could be carried out used the established sex-age equation, so as to apply the MS severity scores of individuals in a clinical context and assess temporal changes in cardiometabolic risk. We add the statements in Discussion. (Lines 231-235)

25) References: please revise and format them properly.

Answer:

References have been revise and format properly.

Reviewer 3 Report

There are many inconsistencies that make not possible to judge on the overall validity of the study.

See my comments attached

Round 2

Reviewer 2 Report

Overall, the authors had addressed all my comments adequately. Just need to polish the English in the introduction and discussion so that it reads better.

Author Response

Thank you for your comment. The article has been edited by a Native English speaker. Attached please find the certification.

Reviewer 3 Report

There are still problems with how FA has been implemented.

1) It is still not clear why 1 factor model was chosen and not 2 or 3and which methods was used to assess the dimensionality of the model. The answer provided does not answer to this fundamental question.

2) Moreover, since different FA models are fitted to different data, loadings for for the different models should be rotated before being compared. It is possible that two or more models are identical, the difference in loadings arising from rotational freedom. I refer the author for an introduction

Rotation in Factor Analysis

Author(s): R. A. Darton

Source: Journal of the Royal Statistical Society. Series D (The Statistician), Vol. 29, No. 3(Sep., 1980), pp. 167-194

Published by: Wiley for the Royal Statistical Society

Stable URL: https://www.jstor.org/stable/2988040

Author Response

Thank you for your comments. Clinically, metabolic syndrome (MS) is composed of and can be diagnosed by five MS components (i.e. waist circumference, FPG, SBP, TG and HDL). The one factor (i.e. MS latent factor) model with an orthogonal rotation was chosen to examine how the various components of MS are correlated with one another and to estimate factor loadings of the five MS components. Models were constructed by adjusting the covariance, and then compared and selected according to the fit measure indices. Our results showed the factor loadings estimated by the optimal model. (Line 127-130) The unstandardized equation of MS severity score can be derived by back-transforming the standardized coefficients and a 5*5 covariance matrix obtained from the factor analysis. (Line 133-134) We address the statement more clearly. The analysis can be performed in the R statistics package (https://www.rdocumentation.org/packages/stats/versions/3.6.2/topics/factanal) and Amos. Some numerical typos are corrected in the revised manuscript. In addition to the references the study cited, the scoring methods have been used and published in past studies (e.g. Gurka et al. 2012. Cardiovascular Diabetology 11: 128; Gurka et al. 2014. Metabolism 63(2); Lee et al. 2017. Metabolism 69; DeBoer et al. 2018. Diabetes Care 41:11; Guo et al. 2018. Diabetology and Metabolic Syndrome 10:42).